# Nanoscale neural network using non-linear spin-wave interference

Ádám Papp[1], Wolfgang Porod [2] & Gyorgy Csaba [1✉]

We demonstrate the design of a neural network hardware, where all neuromorphic computing functions, including signal routing and nonlinear activation are performed by spin-wave propagation and interference. Weights and interconnections of the network are realized by a magnetic-field pattern that is applied on the spin-wave propagating substrate and scatters the spin waves. The interference of the scattered waves creates a mapping between the wave sources and detectors. Training the neural network is equivalent to finding the field pattern that realizes the desired input-output mapping. A custom-built micromagnetic solver, based on the Pytorch machine learning framework, is used to inverse-design the scatterer. We show that the behavior of spin waves transitions from linear to nonlinear interference at high intensities and that its computational power greatly increases in the nonlinear regime. We envision small-scale, compact and low-power neural networks that perform their entire function in the spin-wave domain.

[1] Faculty of Information Technology and Bionics, Pázmány Péter Catholic University, Budapest, Hungary. [2] Center for Nano Science and Technology University of Notre Dame (NDnano), Notre Dame, IN, USA. ✉email: csaba.gyorgy@itk.ppke.hu

The interest in neuromorphic computing hardware increased significantly in recent years, for two main reasons. It was realized long ago that digital systems (let they be CPUs or Graphics Processing Units, GPUs) are rather inefficient for such inherently analog tasks. A more recent development is that traditional, MOS-transistor-based devices turned out to have a strong staying power for Boolean, digital logic—which has driven the research of emerging nanoelectric devices towards neuromorphic, analog problems. These are the application areas where emerging devices have the potential to show substantial benefits over MOS switches[1].

A central challenge of the research on neuromorphic devices is that most computing models require highly interconnected systems, i.e., artificial neurons with a large number of connections, often all-to-all connections. Stand-alone neuronal units have little utility—there should always be an effective way to interconnect those devices to computing systems. This is where wave-based computing concepts show their strengths: if the computing device is realized in a wave-propagating substrate, then interference patterns realize an all-to-all interconnection between points of this substrate.

Recently, Hughes et al.[2] presented a theoretical framework for implementing a recurrent neural network (RNN) in a medium described by a nonlinear wave equation. Specifically, it was shown that if a substrate is described by the nonlinear wave equation and this substrate is excited and probed at given points, then the equations that give the wave dynamics between the prescribed points map to an RNN. In their work, the nonlinearity of the medium is modeled by a spatially varying and intensity-dependent wave propagation speed. Training of the neural network is implemented by adjusting the spatially dependent wave propagation speed by gradient-based computational learning.

The work of Hughes et al.[2] is an original, fresh approach to wave-based computing, but leaves crucial questions unanswered. It is admitted that numerical simulations with the computational learning machine do not fully support the premise of the study, as the RNN-equivalent nonlinear structure shows similar performance to what is achievable by linear propagation. Thus, it is not proved that the presented structure can indeed exploit nonlinear waves to achieve better performance in problems beyond linear signal processing. Furthermore, it is not elaborated, how the form of nonlinearity assumed in the study can be realized by a physical system, albeit a few hints are provided for optical implementations.

The power of wave-based computing has long been harnessed in optical computing and the high interconnectivity is a major selling point for most optical (holographic, interference-based) devices. It is, however, clear that although linear interference is excellent for high interconnections, its computing power is fairly limited. Linear interference is sufficient only for signal processing tasks: general-purpose computing and all variants of neuromorphic computing require some sort of nonlinearity. In optical computing, implementing nonlinearities requires high optical intensities and nonlinearities are often implemented separately from the linear scatterer that provides the interconnections. Other types of waves may implement nonlinear functions in a more natural way. In the present study, we show that spin waves provide both high interconnections and the nonlinearities required for neuromorphic computing.

Spin waves (often also referred to as magnons) are wave-like, collective excitations of a spin ensemble[3]. Here we restrict ourselves to spin waves propagating in ferro- and ferrimagnetic thin films. Spin-wave behavior is approximately linear at low amplitudes, but nonlinearities become significant at moderate intensities. Unlike photons, magnons interact with each other, which is a requirement for non-trivial computation. Spin waves show many similarities to electromagnetic waves and preserve many benefits of optics, e.g., they can maintain long coherence length even at room temperature[4]. Spin waves exist down to sub-100 nm wavelengths at microwave frequencies and they are suitable for integration with electronic components[5].

Spin-wave-based computing devices (which are also referred to as magnonic devices) are widely regarded as a promising beyond-Moore computing paradigm. They have already been experimentally demonstrated for logic, signal processing, and optically inspired computing[3,5]. It has been also realized that high connectivity and built-in nonlinearity make spin waves[6] (and spin-based devices in general[7]) an ideal choice for neuromorphic computing.

However, to actually use spin waves for useful computing tasks, an inverse problem must be solved: one must find a scatterer configuration that yields a certain input/output relation via the formation of an interference pattern. This is in general a daunting task due to the complexity of nonlinear wave propagation.

In the present study, we use the work of Hughes et al.[2] as a starting point, but we study an experimentally realizable magnetic system and model it with full micromagnetic simulations that can precisely describe experimental scenarios. We employ a specific physical system and program it to do true neuromorphic functions. The device is a magnetic thin film, with a spatially non-uniform magnetic field acting on it. A custom micromagnetic solver based on a machine learning framework, Pytorch (https://pytorch.org), is used to design a magnetic-field distribution that steers (scatters) spin waves to achieve the desired function. We named our micromagnetic design engine Spintorch.

Specifically, we design a nanoscale device that performs the functions of a multilayer neural network and which device is physically realizable using microelectronics-compatible technologies. Our work builds on prior art in wave-based computing methods, but rather than choosing an abstract wave equation as its starting point, we start from the physics of magnetic materials. We show that the rich and complex physics of spin waves in a ferrimagnetic thin film can be engineered to perform neuromorphic computation.

For small-amplitude excitations, Spintorch solves an inverse problem for the linear wave equation—it designs a magnetic-field distribution that performs a desired linear operation (such as matrix multiplication, convolution, pattern matching, and spectral analysis, matched filtering) as we will show in "Design of spin-wave scatterers by computational learning." The algorithm has great utility already in this regime, as it automatizes the design of spin-wave-based Radio Frequency (RF) signal processors.

Higher-amplitude spin waves, with a precession angle above few degrees, show nonlinear behavior and Spintorch—the exact same computational learning engine—can be used to design a nonlinear interference device. This device is functionally equivalent to the RNN of Hughes et al.[2] and in "Linear vs. nonlinear interference in the scattering block," we will show how the introduction of nonlinearity increases the computational ability of the device. The spin-wave scatterer becomes a true neural network, exploiting nonlinearity to exceed the performance of linear classifiers.

Our work also addresses the challenge of device integration, possibly the biggest obstacle in the way of practically useful spin-wave computing. There is a number of spin-wave-computing devices designed in the literature, but it is challenging to interconnect such stand-alone magnonic components to functional processor[8]. Our approach circumvents the difficulty of wiring individual nanoscale spin-wave devices: the device and the interconnect are indistinguishable in the wave-computing substrate and they are also designed as one.

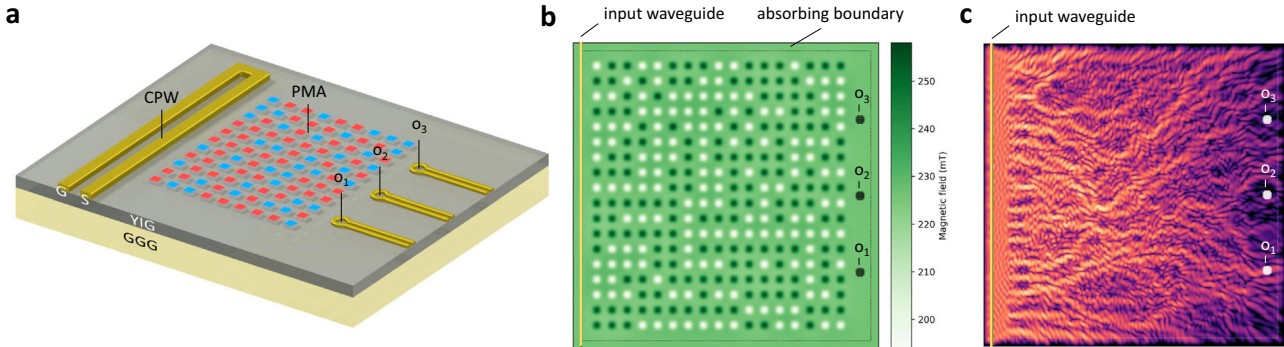

**Fig. 1 Nanomagnet-based spin-wave scatterer. a** The schematics of the envisioned computing device. The input signal is applied on the coplanar waveguide (CPW) on the left and the magnetic state (up/down) of programming magnets on top of the YIG film define the weights. **b** Magnets exhibiting perpendicular magnetic anisotropy are placed on top of the YIG film and generate a bias-field landscape. The training algorithm finds the binary state of the programming magnets. **c** Spin-wave intensity pattern for a particular applied input, which results in a high intensity at $o_1$. The size of the simulation area is 10 μm × 10 μm.

During the finalization of our manuscript, we learned about the results of Wang et al.[9], in which the authors use the inverse-design magnonics to create arbitrary linear, nonlinear, and non-reciprocal devices.

## Results

**Design of spin-wave scatterers by computational learning**. A spin-wave scatterer is a magnetic thin film with spatially non-uniform magnetic field acting on it: this magnetic-field distribution locally changes the dispersion relation of the wave[10], scatters (steers) the spin waves, creating an interference pattern. For the sake of concreteness, we assumed that the wave source is a microwave coplanar waveguide (CPW). The output of the spin-wave scatterer is the spin-wave intensity at particular areas, which experimentally could be picked up via antennas on the film surface.

In order to design experimentally realizable field distributions, a specific geometry of field-generating nanomagnets was assumed, as sketched in Fig. 1a. The punchcard-like pattern of up/down pointing nanomagnets sits on top of a low-damping substrate (such as yttrium iron garnet, YIG) and acts as the program for the spin-wave scatterer. The programming nanomagnets assumed to exhibit strong perpendicular magnetic anisotropy (PMA) and their magnetization is not influenced by the spin waves propagating in the layer underneath—see Supplementary Information for details on the material system. The physical system is straightforwardly realizable; in fact, it is rather similar to the scenarios used in recent experiments, such as in ref. [4]. For some simulations, we used a more fine-grained field distribution (see "Methods" for details).

Spintorch inverse-designs the up/down configuration of the programming magnets in order to realize particular output intensity patterns as a response to an input temporal waveform. The code uses the same gradient-based algorithm that is implemented in Wavetorch (https://pytorch.org; https://github.com/fancompute/wavetorch), but a GPU-based, custom-built full micromagnetic solver is used to model spin-wave propagation as described in "Methods." Instead of using a (nonlinear) wave equation for modeling wave propagation, we solve for the underlying physics by discretizing the modeled region into 25 nm × 25 nm × 25 nm-sized volumes and solve the Landau–Lifshitz–Gilbert (LLG) equations[11] to calculate the precession of magnetic moments in these computational regions. Most importantly, our micromagnetic solver fully accounts for the demagnetizing field and thus the change in magnetic field due to the magnetization precession, which is the source of nonlinearity

in spin-wave propagation. The micromagnetic solver is fully integrated within the computational engine, which performs gradient-based optimization of the trainable parameters finding the optimal up/down magnet configuration.

Micromagnetic simulations, in general, give a highly accurate and predictive description of magnetic behavior, without requiring fitting parameters. To give a recent example, ref. [12] shows how complex interference patterns and nonlinear excitations in a magnetic thin film can be engineered and subsequently measured experimentally. Details about the solver implementation are given in "Modeling the computing substrate."

**Inverse design in the small-amplitude linear regime**. Perhaps the simplest example of inverse design is that of a spectrum analyzer, where the design objective is to focus different spectral components (frequencies) to different spatial locations of the scatterer. In our example, we used a 10 μm × 10 μm scatterer to separate 3, 3.5, and 4 GHz components of the time-domain signal applied on the waveguide. The outputs are 300 nm diameter areas and the time-integrated wave intensity over these areas is defined as the output variable.

The computational learning engine converges to a high-quality design in about 30 training epochs. Here we used small-amplitude spin waves for the training: for precession angles not exceeding a few degrees (excitation fields in the mT range), the computational learning algorithm finds the same solution regardless of the amplitude. The snapshots of Fig. 2 show the spin-wave intensity for the the three frequencies and show that the device performs the required function. The punchcard program that is found by the learning engine is non-intuitive and does not resemble spectrum analyzer designs that were constructed from optical analogies[13]. The field pattern, however, makes a similar impression to refractive index patterns in photonic metamaterial devices[14–16]. The converged scattering pattern also depends on initial conditions that are given to the computational learning engine. The designs, however, all appear to be robust: we verified that switching errors in the magnet states (which are unavoidable in an experimentally realized device) do not affect the performance significantly in most cases.

We would like to point out that the selectivity of the spectrum analyzer design is limited by the relatively small degrees of freedom provided by the ~300 binary values (i.e., the magnets). To scale up the simulations to include more magnets, significantly more computing resources would be needed. Instead, in the following examples, we used external magnetic-field values as training parameters directly without simulating magnets on top.

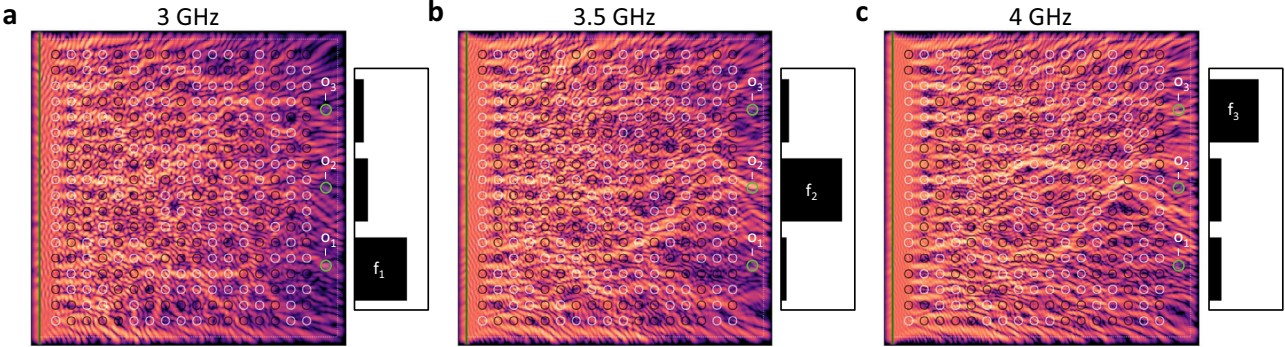

**Fig. 2 Frequency separation by training. a–c** The scatterer was trained to direct frequency components $f_1 = 3$ GHz, $f_2 = 3.5$ GHz, and $f_3 = 4$ GHz to the corresponding outputs denoted by $o_1$, $o_2$, and $o_3$. The bar charts indicate time-integrated intensities measured at the outputs (green circles). The colormaps show time-integrated intensity of spin waves at $t = 30$ ns. Black/white circles are contours of the out-of-plane component of the magnetic field, indicating the state of the magnets on top of the YIG film (same in all cases **a–c**). The size of the simulation area is 10 μm × 10 μm.

This increases the degrees of freedom to ~ 16000 continuous variables, resulting in much better performance at the same computational expense.

The automatized design of linear signal processors alone is an important result and opens many potential application for spin-wave-based devices. Just as photonic metamaterials have much smaller footprint than classical optics devices (such as a 4*f* correlator), the above-designed scatterer (spin-wave metamaterial) has the same advantages over designs based on classical optics (e.g., see refs. [13,17]).

**Vowel recognition in the linear and nonlinear regimes**. For the computational engine, it makes no difference whether the scatterer needs to focus "pure" frequencies to the output points or it has to identify a certain spectral pattern. We tested this by running a vowel-recognition example using the vowel samples available in the Wavetorch package (https://github.com/fancompute/wavetorch)[2]. For comparability, we used the same samples of the data set as in ref. [2], but only the male samples (due to computational resource limitations). In the data set, vowels "ae," "ei," and "iy" were recorded inside the words "had," "hayed," and "heed"; we cropped these samples and only used the middle part of the waveform where the vowel was audible. The waveforms of the vowels were scaled up to microwave frequencies in such a way that the frequency components with significant energy content on the input waveguide launch propagating waves with wavelengths compatible with the scatterer. The scatterer structure was trained to maximize the spin-wave intensity at one of the three output points, which correspond to the recognized vowels. We used four samples of each vowel as a training set. The rest of the 45 samples for each vowel was used as test set.

Some results on the training samples can be seen in Fig. 3a. In 30 training epochs, the system was able to learn to distinguish the vowels "ae," "ei," and "iy," directing the waves toward the correct outputs.

For comparison, we repeated the simulations with increased excitation fields (nonlinear regime, see Fig. 3b). On the training data set, the difference is not significant, although in Fig. 3c, d it is visible that the nonlinear operation achieved better performance and also the convergence is faster. The quality of the vowel-recognition operation is also compared using confusion matrices in case of the testing data set. For three vowels, these are 3 × 3 matrices, where the rows correspond to the predicted output, the columns to the applied input, and the $c_{ij}$ matrix elements give the percentage of cases where vowel $i$ is identified for $j$ vowel as input. For perfect recognition, the confusion matrix is diagonal with 100% at the $c_{ii}$ elements.

At the end of the training, 100% accuracy was achieved on the training data set for both amplitudes, which is largely due to the small size of the training data set (four samples of each vowel). On the larger (41 samples of each vowel) testing data set, the advantage of the the nonlinear operation mode becomes clear. Figure 3e, f show the confusion matrices on the testing data set. The confusion matrix is closer to diagonal in the nonlinear case; it misidentified only 7 out of 123 vowels. The linear device more frequently misidentified all three vowels, but especially "ae," achieving only 60% accuracy.

The confusion matrices in this test scenario characterize the generalization (extrapolation) ability of the network. Based on a very small (four vowels each) learning set, the network had to recognize and classify vowels that it had not seen before. Linear scatterers cannot excel in this job—they match the distinctive spectral features of learned samples, but their ability to generalize from learned data is limited. The nonlinear scatterer appears to behave as a true neural network, which performs nonlinear classification and generalizes (extrapolates) from the training data. We believe that our simulation data may also verify the hypothesis of ref. [2] that nonlinear wave interference acts as an RNN.

We also performed simulations using multiple excitation amplitudes and plotted the corresponding accuracy achieved on the testing data set (Fig. 3g). We observe three distinct regions depending on the amplitude: linear, nonlinear, and chaotic (strongly nonlinear) region. In the linear regime, the accuracy of the system is modest and independent of the amplitude. When the amplitude reaches a certain threshold level, the recognition accuracy suddenly improves significantly and reaches maximum performance around 50 mT excitation field. We attribute this improvement to the emergence of nonlinear effects. However, further increase of the amplitude does not yield to better accuracy due to increasingly chaotic spin-wave dynamics and the system accuracy eventually degrades even below the accuracy of the linear operation.

**Linear vs. nonlinear interference in the scattering block**. Performing successful vowel recognition does not necessarily require a neural network and satisfactory results can be obtained by linear classifiers, as shown in the above example. Using the vowel-recognition example, it is not at all straightforward to identify what benefits could possibly come from using a neural network-like behavior.

However, a fairly simple example can show the computational limitations of linear interference, where the superposition

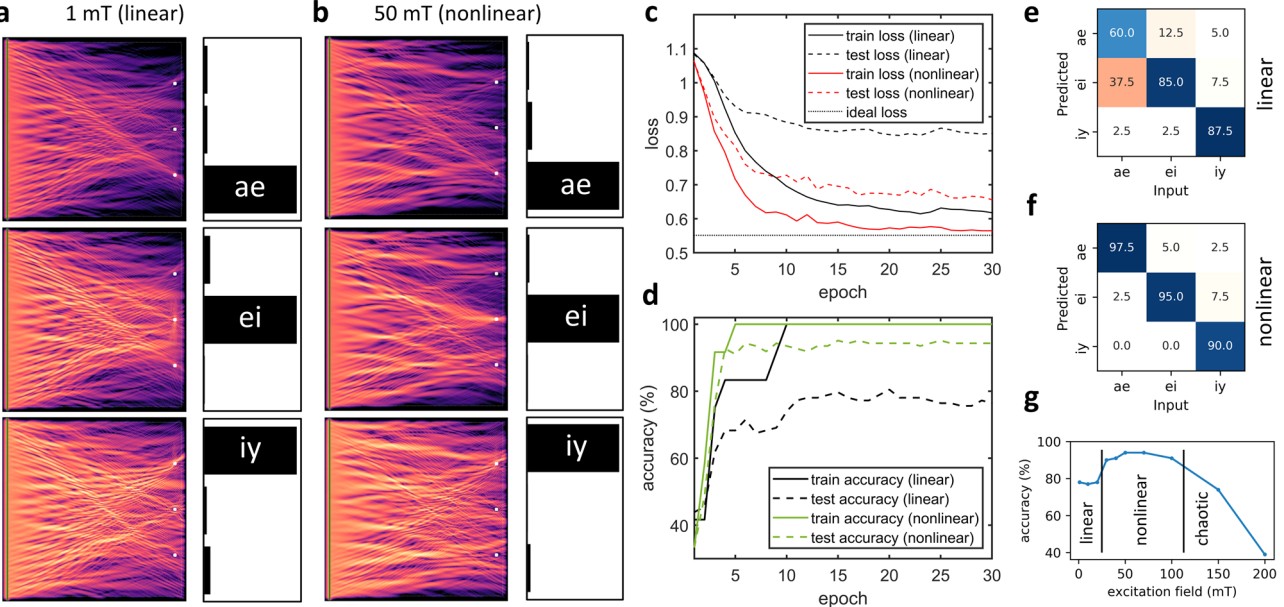

**Fig. 3 Using the spin-wave scatterer for vowel recognition. a, b** Wave intensity patterns, formed in response to the time-domain excitations (vowels). The scatterer was trained to focus waves to the corresponding outputs. The bar charts show the intensity at the output locations (normalized). The linear regime **a** (1 mT excitation field) and the nonlinear regime **b** (50 mT excitation field) performs comparably well on the training data (slight improvement in case of nonlinear waves). **c** Cross-entropy loss decreases during the training, indicating learning. After 30 epochs (training steps), the nonlinear cases achieve better performance compared to the linear case. Note that a nonzero loss value corresponds to the perfect response, indicated by a dotted line. **d** Accuracy of vowel recognition on the training and testing data sets. **e, f** Confusion matrices over the testing data set (123 vowel samples). **g** Accuracy of vowel recognition (test set) as a function of excitation amplitude.

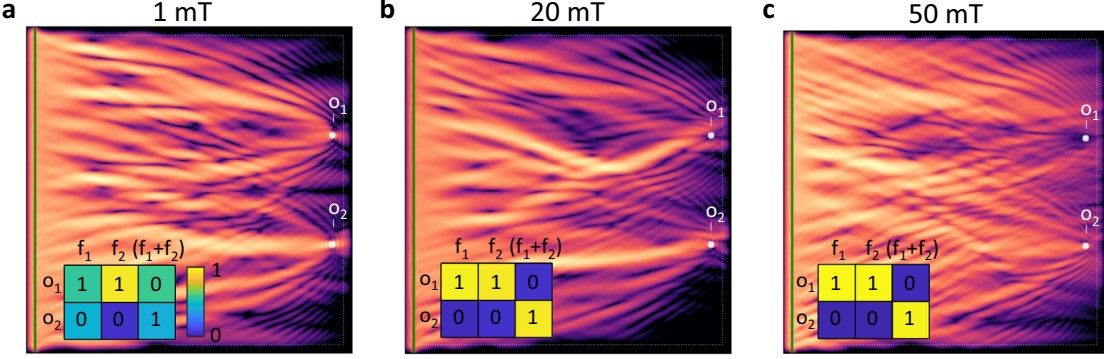

**Fig. 4 A simple example of a problem that is not solvable by a linear system.** Input is encoded in two frequencies ($f_1 = 3$ GHz and $f_2 = 4$ GHz) and the training function is listed in the inset tables (expected results indicated by numbers, output data are shown in color). **a** In the linear case (1 mT excitation field), application of simultaneous frequencies results in both $o_1$ and $o_2$ high (incorrect training). **b, c** In the nonlinear cases, the wave is focused at $o_2$, but $o_1$ is avoided (correct operation). In case of 50 mT excitation, the distinction is even stronger. The colormap shows integrated wave intensity. The size of the simulation area is 10 μm × 10 μm.

principle always holds. In the following example (see Fig. 4), the training goals were to

(A) focus waves on output $o_1$ at 3 GHz input frequency,
(B) focus waves on output $o_1$ at 4 GHz input frequency, and
(C) focus waves on output $o_2$ if and only if 3 and 4 GHz are simultaneously present.

Clearly, condition (C) is inconsistent with the superposition of (A) and (B).

We used excitation amplitudes in the linear (1 mT) and moderately nonlinear (20 and 50 mT) regimes, and run the training for 30 epochs in every case. The resulting spin-wave intensity snapshots are shown in Fig. 4. It is clearly visible that the results of the training are different in case of different amplitudes. The paths traveled by the waves are completely different in the

three cases. It is also clear from the snapshots that the linear case failed to focus on $o_2$, whereas the nonlinear cases were clearly focusing to the bottom output ($o_2$) avoiding $o_1$.

As expected, the linear case could not provide the desired outcome: the output of the two-frequency case is a linear combination of the outputs observed with single-frequency excitations. On the contrary, the operation in the nonlinear regime achieved good results, with the highest amplitude excitation giving the best outcome. Quantitatively, the loss function, which quantifies the quality of the computational learning, yields the same conclusion: the linear case did not show a convergence over the 30 epochs, whereas the nonlinear cases converged to an optimal loss value. The highest excitation amplitude achieved lower loss at the end of the training and its convergence was also faster.

This elementary example demonstrates a crucial difference between the computational power of linear and nonlinear spin waves—and this difference is expected to manifest itself for more complex operations, such as the high-amplitude vowel-recognition example in "Vowel recognition in the linear and nonlinear regimes." It also serves as a proof that our computation engine is able to exploit the nonlinearity of spin waves.

## Discussion

Nowadays complex neuromorphic computing pipelines are implemented on CPUs and GPUs, which posses high computing power, but have poor energy efficiency for analog neuromorphic tasks: computing steps are implemented on digital (often floating point) arithmetic and each of those steps consume energy in the $E = 10^{-11}$ J range. A typical processing computing primitive (such as a single convolution with smaller precision in convolutional neural network) consumes roughly the same amount.

Spin waves in nanoscale magnetic structures carry little energy: the total magnetic energy stored in the patterns of say Fig. 4c is about $E \approx 1000$ eV, $E \approx 10^{-16}$ J. Patterns in the linear regime hold orders of magnitude less energy, in the few-eV range. The stored energy can serve as a first estimate of the energy that is dissipated in the magnetic domain in each neural computation step. The time it takes for the interference pattern to build up is on the order of $t = 10$ ns; thus, the spin-wave scatterer simultaneously achieves low power and high speed. These estimations represent orders of magnitude improvement when compared to the above-mentioned energy of approximate convolution or floating point operation, indicating great potential for the spin-wave-based processor. Neurons and synapses based on other emerging devices[18] also consume significantly more energy than the stated $E \approx 10^{-16}$ and they do it at slower computational times. The neural operation done by the scattering block in the nonlinear regime is also considerably more complex than a convolution or what is performed by the synapses and neurons in ref. [18].

The power dissipated in the magnetic domain is just a lower bound for net power consumption. The spin-wave scatterer may find application as a hardware accelerator in electrical circuitry—and, in that case, the net energy efficiency of spin-wave-based computing block is dominated by the magneto-electric transducer. More specifically, picking up magnetic oscillations from sub-square-micrometer areas will induce less than a microvolt voltage in the transducer antenna, possibly even less than that. Amplifying such small and high-frequency signals requires significant microwave circuitry, which consumes at least 10 mW of power[19,20]. Assuming a GHz date rate, this gives $E = 10^{-11}$ J per output point. Transduction on the input side (creation of spin waves) is less challenging as can be done by acceptable efficiency using CPWs and a single waveguide can excite a larger number of scatterers. It is worth noting that the computational learning engine can implement any kind of readout mechanism; thus, any shape, size, and type of detector may be used, depending on the application. For example, inverse spin-Hall effect[21] might be used to detect directly the amplitude of spin waves, whereas the detector size is not limited by the spin-wave wavelength.

The net power efficiency of the spin-wave scatterer is comparable to that of electronic implementation for a simple operation (i.e., a convolution). If large internal complexity can be reached in the scatterer with a single or very few inputs, then the spin-wave scatterer potentially leads to several orders of magnitude performance gain compared to electronic implementations.

Optical reservoir computing[22,23]—another promising hardware for accelerating neural computations—consumes on the order of $E = 10^{-11}$ to $E = 10^{-12}$, which is comparable to a small spin-wave scatterer with I/O, but it comes with a significantly larger device footprint. Strictly linear operations in optics may be performed with much higher energy efficiency (due to the more straightforward scalability of optical systems[24]), but such systems require several additional components for general-purpose computation.

Most likely, the capabilities of our spin-wave scatterer are limited by the computational learning method we use to design it. Machine learning on the micromagnetic model is computationally intensive (see "Implementation of the computational learning engine") and we could not design larger systems than the ones presented here and compare those to state-of-the-art neural networks.

Spin waves are a leading candidate for non-electrical information processing and magnonic devices have been designed for many different purposes, such as Boolean logic gates[25], and signal processors[13]. In many cases, magnonic computers are derived from photonic computing devices and, most often, classical photonics is used as an inspiration, with lenses, mirrors, and interferometers designed in the spin-wave domain[26].

Our work advances the state-of-the-art of magnonic computing devices on two fronts. First, we demonstrated that the computational tools developed for the inverse design of photonic metamaterials (a.k.a. photonic inverse design) can be applied in the spin-wave domain: convolvers, spectrum analyzers, matched filters, and possibly a large variety of RF signal processing devices can be designed in a fully automatic way. Spin waves, unlike electromagnetic waves, seamlessly transition to a nonlinear regime at higher excitation amplitudes. Apparently, the computational design algorithm operates just as well if the underlying wave propagation is a nonlinear wave and designs devices based on nonlinear interference. The second and perhaps the most important result of our work is that the capabilities of such-designed nonlinear interference devices go beyond linear signal processing and they are likely equivalent to RNNs. The device realizes all the interconnections, weighted sums, and the nonlinearities in a single magnetic film. The techniques we demonstrate pave the way to magnonic (spin-wave) devices way beyond the complexity of stand-alone logic gates or elementary signal processors.

Wave-based general-purpose computing—and more generally, computing in a material substrate by the laws of physics—is a longtime dream of the emerging computing community[27–30]. Possibly, spin-wave-based nonlinear processors enable practical, physical realization of these ideas and bring closer to the fulfillment of this vision.

## Methods

**Implementation of the computational learning engine.** Spintorch is a modified version of Wavetorch (https://github.com/fancompute/wavetorch), in which we implemented a full micromagnetic solver to precisely model spin-wave behavior. The numerical engine for inverse design is built on the popular and open-source machine learning framework, Pytorch (https://pytorch.org). An important feature of Pytorch is the automatic gradient calculation, which allows automatic backpropagation throughout complicated multilayered computational flows. In our system, this means gradient calculation can be performed backwards in time throughout the whole wave propagation. This allows us to perform gradient-based optimization of the trainable parameters, e.g., the applied magnetic-field distribution. Pytorch also provides a number of optimizers, loss functions, and data-loading modules, so we did not need to implement these from scratch. Pytorch modules can run on CPU or GPU (using the CUDA programming interface), without any device-specific coding on the user side.

In order to exploit the automatic gradient calculation feature of Pytorch, custom modules must use the internal methods for implementing the forward path of the system. This way, the backward method is automatically generated on the fly by building a computational graph and saving the required intermediate results. Thus, readily available micromagnetic solvers (such as OOMMF or mumax3[11]) cannot be integrated in Pytorch, because these do not build a computational graph and do not save intermediate results for backpropagation.

Due to the requirement of saving all (micromagnetic) computational steps, the algorithm requires a large amount memory to run—the size of the systems we could simulate is limited by the available GPU memory. Due to this limitation, we could not test larger systems and much larger data sets.

**Modeling the computing substrate.** The dynamics of the magnetic media are described by the LLG equation and takes into account all relevant physical

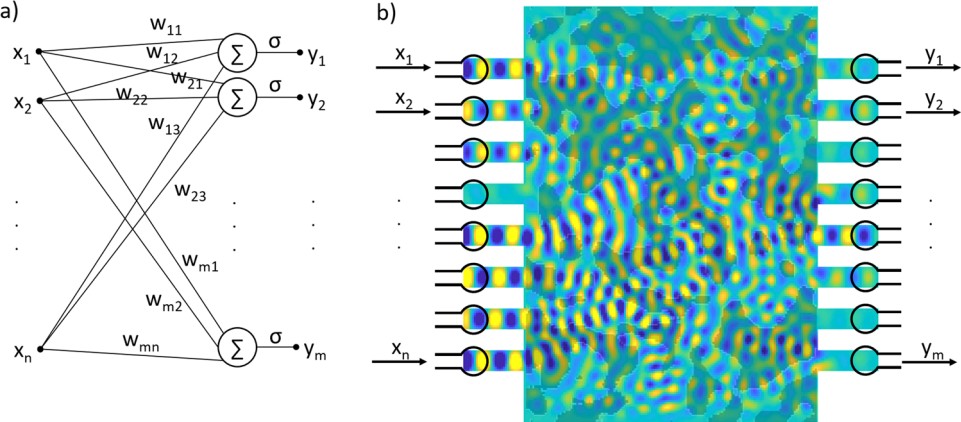

**Fig. 5 Understanding the spin-wave scatterer as a neural network. a** Is a schematic of an *n*-input, *m*-output perceptron layer (**b**) is an *n*-input, *m*-output spin-wave scatterer.

interactions in the micromagnetic model. Elementary magnetic moments are represented by three-dimensional vectors and we use a finite difference discretization with a rectangular grid. The dynamics of magnetic moments depend on the torque exerted on them by the effective magnetic field, which is a sum of several field components. Most importantly, it includes an (space-, and time-dependent) external field, the dipole fields of other magnetic moments, and the exchange interaction between neighboring volumes. The external field contains a bias field, the time-dependent excitation field, and also the field coming from the PMA magnets (which is optimized by machine learning). We neglect any dynamics of the PMA magnets (e.g., due to coupling of spin waves from YIG), as these fields are orders of magnitude lower[10]. The dipole interaction is a long-range effect; thus, it is the most computationally expensive part of the calculation. We used Fast Fourier transform (FFT)-based acceleration for calculating the solution of the Poisson equation (i.e., determine the dipole fields), for which the GPU-accelerated FFT module of Pytorch enabled effective implementation. The exchange field is calculated only between nearest neighbors, as exchange field is local. Exchange-field calculation is implemented using a convolution with a Laplacian kernel. The time stepping of the differential equation is realized by a standard fourth-order Runge–Kutta method. The LLG equation also includes a damping term, which is implemented in our model; thus, realistic attenuation of spin waves is simulated. A similar model (i.e., implemented in Matlab and does not use computational learning) is described in ref. [31]. Damping is also used to realize absorbing boundary conditions, so we could accurately model a few-micrometer-sized, finite region of an extended magnetic film.

The micromagnetic model fully accounts for the nonlinearities appearing at higher intensities: these nonlinearities are a direct consequence of the dependence of the demagnetizing field on the local magnetization and the spin-wave amplitude. A detailed description of the dispersion relation and nonlinearity of spin waves can be found in ref. [32]. The nonlinear wave equation for spin waves cannot be stated in a simple form, as the source of nonlinearity is the amplitude-dependent magnetic field, so the nonlinearity enters into the dynamic equations via Maxwell's equations. Experimental demonstration of chaotic behavior of spin waves due to modulational instability at high-amplitude levels can be found in ref. [33]. In this work, we used moderate amplitudes to avoid chaotic behavior, which degrades the ability of the device to perform the desired functionality.

We verified our solver by comparing results with the widely used mumax3 solver[11]. The high-level use of GPU-based functions and the overhead of automatic gradient computation makes our code less efficient as a general-purpose micromagnetic solver; however still, running times are comparably fast and more than 100,000 cells with a few thousand timesteps can be simulated in minutes on a state-of-the-art GPU. This makes it possible to embed the solver into the learning algorithm and train the system with multiple samples and epochs within a few hours or, with a larger training set, days.

**Magnetic material properties**. YIG is used as a medium for low-damping spin-wave propagation and arrays of nanomagnets with PMA provide control over spin waves via their dipole fields. PMA magnets are bistable (magnetization pointing either upwards or downwards) if their size is below the single domain limit (typically less than a few hundred nanometers).

Such a system could provide a reconfigurable means to programming spin-wave-based neural networks, by individual switching of the nanomagnets. This implementation of the scatterer shows many benefits over a lithographically patterned (hardwired) scatterer. In our model, we included the calculation of realistic dipole fields of the nanomagnet arrays, which works for any configuration.

The chosen material system and geometry is just one of many possible choices. Metallic ferromagnets could have been used in place of the YIG film—these have higher damping (shorter propagation length), but easier to integrate and access

electrically. In addition, instead of the stray-field programming, lithographically defined patterns (lithography followed by etching) could have defined the function of the scatterer. Fine-grained tuning of YIG magnetic properties can be achieved by Focused Ion Beam (FIB) irradiation of a YIG film[34,35] that continuously changes magnetic parameters as a function of the local dose. We expect that our computational engine can be used with similar effectiveness when film magnetic parameters are adjusted by training, instead of designing the applied field pattern as we have done in this work.

**Linear spin-wave scatterer as a perceptron layer**. Here we would like to show how a spin-wave scatterer block can represent a single layer of a neural network (perceptron layer). A perceptron layer can be described mathematically as a linear transformation (vector–matrix multiplication) followed by a nonlinear activation function: $\mathbf{y} = \sigma(\mathbf{Wx})$, where $\mathbf{x}$ is a vector of length $n$ representing the input, $\mathbf{W}$ is an $m \times n$ matrix that contains the trainable weights of the perceptron layer, and $\sigma$ is an activation function applied on every output channel (in the simplest case, a threshold function). Regarding its functionality, a perceptron performs a linear classification, so a layer of perceptrons performs $m$ different linear classifications. The linear transformation ($\mathbf{W}$) can be performed by a spin-wave scatterer block, as depicted in Fig. 5. If the amplitude of the spin waves is sufficiently small, the wave propagation can be described by the linear wave equation. Input signals routed to input antennas generate spin waves with corresponding amplitudes and phases. Waves travel through a region where the effective refractive index is spatially varying according to the program (the desired linear transformation). The wave intensity from every input will be distributed among the outputs by the scatterer map (some losses may also occur). As the wave propagation is assumed to be linear, the activation function has to be implemented in the readout circuitry.

The matrix representation of a given scatterer map can be constructed using the superposition principle: exciting the inputs one at a time with unit amplitude (base vectors) and recording the outputs will produce the columns of the equivalent matrix. The inverse problem is, however, more cumbersome to solve in general. One possible approach is the machine learning method described by Hughes et al.[2], which is directly applicable to any system that obeys the linear wave equation or can be modified for nonlinear equations.

Such a device, apart from dynamic range and scaling limitations, can in principle realize any perceptron layer. However, the computing capabilities of a single layer are limited to linear classification and even some relatively simple operations (such as an XOR gate) is impossible to realize using this device. To overcome such limitations, one could create a multilayer neural network using such devices sequentially, but any advantages that come from the low-power operation and compactness of the spin-wave scatterer would be overshadowed by the required readout circuitry. Any approach to exploit the benefits of the highly interconnected nature of wave interference should minimize the number of input–output conversions. Thus, we investigated the feasibility of exploiting the nonlinearity of spin waves, which would allow implementing a multilayer neural network (or an RNN) within a single scattering block.

Compared to the linear operation, nonlinear wave propagation implements a "distributed activation function"; thus, the spin-wave substrate is more akin to multilayer neural networks. Moreover, due to back-scattering of spin waves, loops can form in the scatterer that resemble the operation of RNNs. Due to the delay of wave propagation, such loops could also implement a fading memory, which is not exploited in the current investigation, but provide additional potential for the spin-wave substrate. We briefly note that the activation function in the system is bounded in the [−1,1] interval, which is shown to be a satisfactory condition in multilayer feedforward networks for universality[36]. Although investigation of the limits to the computational capability of the spin-wave substrate is out of the scope of this study, in principle, the presented substrate could approximate arbitrary functions.

We finally note that our study is not the first to describe physically realizable wave-computing systems: there are other proposals (e.g., see ref. [37]) that even experimentally demonstrate wave-based computing protocols. However, to our knowledge, our study is the first to show an on-chip, integration-friendly wave-computing device where nonlinearities play a key role.

## Data availability

The Spintorch package is available from the authors upon request. A publicly available version is planned for release in 2021.

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

## Acknowledgements

We are grateful for fruitful discussions and encouragement from Markus Becerer (Technische Universität München), Andrii Chumak, Qi Wang (University of Vienna), and Philipp Pirro (TU Kaiserslautern). This research was in part financially supported by the DARPA Nature as Computer (NAC) program and we are grateful to the DARPA team for helpful advice. Á.P. received funding from the postdoctoral grant (PPD-2019) of the Hungarian Academy of Sciences.

## Author contributions

G.C. and W.P. conceived the original idea. Á.P. designed the computational engine, performed the micromagnetic simulations, and explored the role of nonlinearities. Á.P., W.P., and G.C. wrote the manuscript. All authors discussed the results and reviewed the manuscript.

## Competing interests

The authors declare no competing interests.
