## [Peer Review File · Nature Communications]

Reviewers' Comments:

Reviewer #1:

Remarks to the Author:

Comments

In this work, the authors propose a spin wave-based neural network, which can perform various neuromorphic computing functions, such as spectrum analyzer and vowel recognition. Their results show that such-designed device can indeed exploit nonlinear waves to achieve better performance in problems beyond linear signal processing. The idea of using the propagation and interference of spin wave for neuromorphic computing is interesting and novel, and this topic is promising for a new class of analog machine learning platforms. In general, the manuscript can be accepted for publication after the following issues are clarified and corrected.

1) Fig.3 demonstrates the cross-entropy loss of training sets, the confusion matrices for the testing set in linear and nonlinear cases, respectively. What about the cross-entropy loss of testing sets and the confusion matrices of the training sets? Most importantly, the prediction accuracy as a function of training epochs over the testing and training datasets should be given as shown in [Tyler W. Hughes et al., *Sci. Adv.* **5**, 6946 (2019)]. The authors claim that the confusion matrices on the training data set are perfectly diagonal for all amplitudes due to the small training sets and the comparably large internal complexity of the scatterer. It might be true to some extent. In my opinion, it is necessary to show above mentioned elements to draw more solid conclusions and prove that there is no overfitting problem. In addition, please give more details of the training dataset and testing dataset in one paragraph. Ref. [5] (Science Advances 5, no. 12 (2019): eaay6946.) in their references clearly explained the training details as follows. Please follow their example to give a more clear description.

Training a physical system to classify vowels

We now demonstrate how the dynamics of the wave equation can be trained to classify vowels through the construction of an inhomogeneous material distribution. For this task, we use a dataset consisting of 930 raw audio recordings of 10 vowel classes from 45 different male speakers and 48 different female speakers (21). For our learning task, we select a subset of 279 recordings corresponding to three vowel classes, represented by the vowel sounds ae, ei, and iy, as contained in the words had, hayed, and heed, respectively (Fig. 2A).

- 2) In section 2.3, they think the operation in the nonlinear regime achieved good results, with the highest amplitude excitation giving the best outcome. Is it possible to plot the relationship between the outcome (such as the final loss and the prediction accuracy) and the excitation amplitude? Is there a saturation point? Or is there a maximum value limit for the excitation amplitude?
- 3) The authors mention that the linear and nonlinear wave equations several times. Is it possible to provide the corresponding theoretical formulas or representative references, which can give readers a clearer physical scenario.
- 4) To improve the significance of this work, I would like the authors to make a comparison between the proposed device with the conventional RNN in terms of the prediction accuracy.
- 5) In the main text, the authors claim that the programming nanomagnets assumed to exhibit strong perpendicular magnetic anisotropy (PMA). However, in section 5.2 (e.g. methods), the effective magnetic field is just described by the sum of the external field, the dipole field and exchange interaction. I wonder why the PMA field is

missing here.

Reviewer #2:

Remarks to the Author:

This paper demonstrated a design of neural networks based on a reconfigurable inhomogeneous magnetic system, through which the propagation and interference of spin wave can be trained to realize neuromorphic computing functions such as vowel recognition. The content of the paper is based on the theoretical framework proposed by Hughs et al [Ref. 5], the authors proposed a particular wave-based material system - the magnetic system with spin wave for realizing this framework. In particular, the spin wave possess intrinsic non-linearity, which is crucial for neuromorphic purposes. The authors use micromagnetic simulations in combine with machine learning technique to demonstrate the feasibility of the proposed magnetic neural network and its estimated the energy and device footprint, which is significantly lower or comparable to the existing optical reservoir computing hardwares.

Overall, this is a nice piece of work for showing a realizable platform for neuromorphic computing. However, the main concern of this reviewer about the novelty of the paper is the following: what's the major step forward beyond the theoretical proposal given in Ref. 5. A simple physical realization of the proposal does not seem to be suffice. Therefore, I cannot recommend the paper for publication before the authors can provide convincing argument about the major advances on top of earlier works.

Some other comments:

1) The output is defined as time-integrated wave intensity over certain area. Is the intensity defined as the square of the spin wave amplitude? If so, how to pick up such time-integrated signal in reality? And how crucial is the size of the area?

2) The magnetic dots with perpendicular anisotropy placed on top of the magnetic YIG film served as trainable weights. During the simulation, I assume the authors only simulated the magnetic dynamics within the YIG film and the magnetic dots are assumed be to static. But in reality, the magnetization in the magnetic dots would also have dynamics. Will or will not the dynamic motion in the magnetic dots alter the results, and why?

3) Fig. 3 shows the intensity of the wave have significant influence on the results, i.e. the non-linear spin wave enhances the vowel-recognition rate. What's the principle behind such enhancement?

Reviewer #3:

Remarks to the Author:

The authors numerically study the propagation of spin waves in a scattering medium. The authors train the scattering medium to perform some computations. Specifically, they focus on frequency-dependent scattering to analyze vowel data, including nonlinearity in the evolution.

The results are not particularly surprising, given the many recent papers on computing by waves (e.g., PRX 8, 041037 (2018)). For this reason, I am not supporting the acceptance by NCOMM.

The main drawback is the absence of experimental results, which may eventually boost the impact of the paper. It is well established that waves compute by proper training of the medium, irrespectively of the type of waves (as outlined in ref. 5). The fact the authors report some simulations with spin waves is not unexpected or exciting.

Another major drawback is that the authors do not test their simulation with a standard dataset for benchmarking machine learning methods, as the MNIST.

Hence, I am against the acceptance of the manuscript.

Minor comments follow

- I do not like the general style of the paper; many vague statements and colloquial and verbose style ("skyrocketed," "stellar numbers," "soft-nonlinear," "is less of a problem," among others)

- The authors vaguely remark the rule of nonlinearity. The may outcome is that the nonlinearity improves training. This conclusion is vague, not only because the training and testing datasets are absent but also because the authors do not discuss the model universality. Non-polynomial nonlinearity is a needed ingredient for universality, and no insight is given in this direction.

General remarks to all the reviews:

Please find below our responses to the points raised by the two reviewers. We copied their comments (shown in green), our replies are shown in black, and we show revisions in blue, as they appear in our revised manuscript. A marked-up copy of the manuscript is also attached, which shows the differences between the originally submitted and the revised copy.

After reading the reviews and carefully re-reading our paper we realized that the significance of our results for the magnonics (spin-wave computing) community should have been emphasized much stronger. The initially-submitted version of the paper made the impression that it is primarily a computational learning study. The reviewers raised concerns, that our results serve just as a case study for wave-computing concepts, which have recently appeared in the literature. This is not the case: our paper has the promise to open new avenues for the design of real physical magnonic devices with real-world applications. The techniques we demonstrate enable the magnonic (spin-wave device) community to design devices very much beyond the complexity of individual logic gates or simple signal processors.

Our work addresses (possibly solves) a much sought-after problem in magnonic device research. We demonstrate the design of complex processing units that do not require additional interconnections, or auxiliary electrical components (besides inputs and outputs) to function. While stand-alone devices (logic gates, signal processors) are intensely researched by the community, interconnecting these units to functional components is challenging. Our approach circumvents this problem by designing the device and the interconnections at once, realizing the desired input-output scenario.

We attempted to emphasize this important point stronger in the revised paper. We thank you for the reviewers for pointing out the above the issues – we think that our revised paper communicates much better our message and clarifies the missing details.

Reviewer 1

In this work, the authors propose a spin wave-based neural network, which can perform various neuromorphic computing functions, such as spectrum analyzer and vowel recognition. Their results show that such-designed device can indeed exploit nonlinear waves to achieve better performance in problems beyond linear signal processing. The idea of using the propagation and interference of spin wave for neuromorphic computing is interesting and novel, and this topic is promising for a new class of analog machine learning platforms. In general, the manuscript can be accepted for publication after the following issues are clarified and corrected.

1) Fig. 3 demonstrates the cross-entropy loss of training sets, the confusion matrices for the testing set in linear and nonlinear cases, respectively. What about the cross-entropy loss of testing sets and the confusion matrices of the training sets? Most importantly, the prediction accuracy as a function of training epochs over the testing and training datasets should be given as shown in [Tyler W. Hughes et al., Sci. Adv. 5, 6946 (2019)]. The authors claim that the confusion matrices on the training data set are perfectly diagonal for all amplitudes due to the small training sets and the comparably large internal complexity of the scatterer. It might be true to some extent. In my opinion, it is necessary to show above mentioned elements to draw more solid conclusions and prove that there is no overfitting problem. In addition, please give more details of the training dataset and testing dataset in one paragraph. Ref. [5] (Science Advances 5, no. 12 (2019): eaay6946.) in their references clearly explained the training details as follows. Please follow their example to give a more clear description.

We updated Fig. 3. according to the suggestions of the Reviewer, adding new data from new simulations. Figure 3c now includes the loss during the training epochs for both the training and testing data sets in case of both the linear/nonlinear operation. Similarly, we calculated prediction accuracy for every training epoch

and plotted the test/train accuracy for both the linear and nonlinear cases (Fig. 3d). The confusion matrices for the training data set are perfectly diagonal (as we indicated in the text before), and this fact is clearly visible now in the accuracy of the curves (in Fig. 3d accuracy reaches 100 % very quickly on training dataset for both the linear and nonlinear cases). In light of this, we did not display the perfectly diagonal confusion matrices for the training sets, since they do not contain additional information.

We now included more information about the data set. We also emphasize that we used the same data set as Ref [5] for better comparability, and thus we also refer to their description of the data set.

Here is how we incorporated this discussion in the revised version:

Updated Fig 3. and figure captions (see in the manuscript).

“For comparability, we used the same samples of the dataset as [5] but using only the male samples (due to computational resource limitations). In the dataset vowels ‘ae’, ‘ei’, and ‘iy’ were recorded inside the words ‘had’, ‘hayed’, and ‘heed’, we cropped these samples and only used the middle part of the waveform where the vowel was audible.

(...)

We used two or four samples of each vowel as a training set. The rest of the 45 samples for each vowel was used as test set.”

2) In section 2.3, they think the operation in the nonlinear regime achieved good results, with the highest amplitude excitation giving the best outcome. Is it possible to plot the relationship between the outcome (such as the final loss and the prediction accuracy) and the excitation amplitude? Is there a saturation point? Or is there a maximum value limit for the excitation amplitude?

We performed additional simulations on the vowel recognition device with multiple excitation amplitudes. The accuracy of the device on the testing set as a function of excitation amplitude is now depicted in Fig. 3g. We point out three separate amplitude regions: in the linear region, the performance of the device is independent of the amplitude. As the amplitude increases into the nonlinear region (after a sudden transition) the performance of the device improves significantly. Further increase of the amplitude gradually hinders the performance of the device, which we attribute to the onset of chaotic behavior of spin waves (modulational instability). We believe these three regions are fundamental to spin waves and similar behavior can be expected for other training tasks as well (as in case of the one described in Section 2.3). We now added a paragraph about these results at the end of Section 2.2.

Here is how we incorporated this discussion in the revised version:

“We also performed simulations using multiple excitation amplitudes and plotted the corresponding accuracy achieved on the testing data set (Fig. 3g). We observe three distinct regions depending on the amplitude: linear, nonlinear, and chaotic (strongly nonlinear) region. In the linear regime the accuracy of the system is modest, and independent of the amplitude. When the amplitude reaches a certain threshold level, the recognition accuracy suddenly improves significantly, and reaches maximum performance around 50~mT excitation field. We attribute this improvement to the emergence of nonlinear effects. However, further increase of the amplitude does not yield to better accuracy due to increasingly chaotic spin-wave dynamics, and the system accuracy eventually degrades even below the accuracy of the linear operation.”

3) The authors mention that the linear and nonlinear wave equations several times. Is it possible to provide the corresponding theoretical formulas or representative references, which can give readers a clearer physical scenario.

We now included references in Section 5.2 that give a better description of the physical scenario. We would like to point out that the nonlinear behavior of spin waves is more complicated to handle analytically than the nonlinear wave equation considered in Ref [5], thus we have to completely rely on numerical methods.

We also added a brief explanation about the physical background of the nonlinearity: magnetization dynamics depends on the local magnetic field, but the local magnetic fields depend also on the amplitude of the magnetization oscillations. We do not state the full system of equations, but mention this point in the Methods section.

Here is how we incorporated this discussion in the revised version:

“A detailed description of the dispersion relation and nonlinearity of spin waves can be found in [30]. The nonlinear wave equation for spin waves cannot be stated in a simple form as the source of nonlinearity is the amplitude dependent magnetic field, so the nonlinearity enters to the dynamic equations via Maxwell's equations. Experimental demonstration of chaotic behavior of spin waves due to modulational instability at very high amplitude levels can be found in [31]. In this work we used moderate amplitudes to avoid chaotic behavior, which degrades the ability of the device to perform the desired functionality.”

[30] Wu, Mingzhong. "Nonlinear spin waves in magnetic film feedback rings." *Solid State Physics* 62 (2010): 163-224.

[31] Wu, Mingzhong, et al. "Excitation of chaotic spin waves through modulational instability." *Physical review letters* 102.23(2009): 237203

4) To improve the significance of this work, I would like the authors to make a comparison between the proposed device with the conventional RNN in terms of the prediction accuracy.

The focus of our study is the investigation of a physical substrate that could perform computing tasks, and the design (training) of such a substrate. Our primary goal was to show that a spin-wave scatterer can perform computing tasks and nonlinear scattering increases the complexity of the computing problems. We are simulating a small system with all the micromagnetic interactions accurately modeled, which is a computationally intensive task. Benchmarking such a simulated physical system against an artificial neural network that is directly realized in software would, in our opinion, not be a meaningful and fair comparison. Our model system is very small compared to most systems discussed in the literature of Artificial Neural Networks. The reason for this is that is the amount of computational resources it requires, since we need to simulate a complex physical system. This places a limitation on the complexity of the problems we can currently tackle, and also the size of the training and testing data sets. Of course, benchmarking a physically realized system would be a meaningful comparison, as the physical substrate would operate real-time on the data, without the simulation overhead.

We used the vowel recognition problem to be comparable to the results of Ref [5], and we believe that we succeeded in demonstrating that nonlinear spin-wave behavior has greater computational complexity than what is possible using linear wave interference (in this respect the simulation results of Ref [5] were inconclusive). Before attempting a meaningful benchmark against well-established methods, we need to scale up the system to be able to solve problems with practical significance. We don't see any fundamental limitation in this sense, but the efficiency of our training algorithm needs to be improved so that we could apply it to larger-scale problems. We also plan to work on experimental verification of the presented principles, which would allow us to design a more realistic system, and better estimation on the limitations of the device.

We can more confidently make a comparison between electrical and spin-wave hardware. The spin-wave interference formation process takes a couple of nanoseconds in time and the size of our entire computing substrate is 10 micrometers – a digital or analog electronic circuitry performing the same RNN task would likely consist of hundreds (possibly thousands) of transistors, with significantly higher power consumption, significant routing complexity and higher footprint. These points are mentioned in the benchmarks section.

Here is how we incorporated this discussion in the revised version (just before concluding the paper):

“The computational learning method we use is computationally intensive (see Section 5.1) and we could not design larger systems than the ones presented here and compare those to state of art neural networks.”

5) In the main text, the authors claim that the programming nanomagnets assumed to exhibit strong perpendicular magnetic anisotropy (PMA). However, in section 5.2 (e.g. methods), the effective magnetic field is just described by the sum of the external field, the dipole field and exchange interaction. I wonder why the PMA field is missing here.

The magnetic system consists of two parts: a YIG thin film and Co|Pt nanomagnets on top. PMA is present in Co|Pt (and similar) multilayer systems, but not in YIG. In the simulations we assume that PMA keeps the nanomagnets magnetized completely in out-of-plane direction, and the static stray field of such magnets is possible to calculate analytically. Thus, in the dynamic micromagnetic simulations the PMA magnets themselves are not present, only their stray fields are included.

To make things clearer, we added a sentence stating that physically the ‘programming’ field is included in an external field component:

“The external field contains a bias field, the time-dependent excitation field, and also the field coming from the PMA magnets (which is optimized by machine learning). We neglect any dynamics of the PMA magnets (e.g. due to coupling of spin waves from YIG) as these fields are orders of magnitude lower [9].”

Reviewer 2

This paper demonstrated a design of neural networks based on a reconfigurable inhomogeneous magnetic system, through which the propagation and interference of spin wave can be trained to realize neuromorphic computing functions such as vowel recognition. The content of the paper is based on the theoretical framework proposed by Hughs et al [Ref. 5], the authors proposed a particular wave-based material system - the magnetic system with spin wave for realizing this framework. In particular, the spin wave possess intrinsic non-linearity, which is crucial for neuromorphic purposes. The authors use micromagnetic simulations in combine with machine learning technique to demonstrate the feasibility of the proposed magnetic neural network and its estimated the energy and device footprint, which is significantly lower or comparable to the existing optical reservoir computing hardwares.

Overall, this is a nice piece of work for showing a realizable platform for neuromorphic computing. However, the main concern of this reviewer about the novelty of the paper is the following: what’s the major step forward beyond the theoretical proposal given in Ref. 5. A simple physical realization of the proposal does not seem to be suffice. Therefore, I cannot recommend the paper for publication before the authors can provide convincing argument about the major advances on top of earlier works.

We think that the approach of Ref. 5 is genuine, and it was a great inspiration for our work, we employed part of their theoretical framework and we also built on their algorithmic approach (the way of using gradient based training for inverse design). We believe that there are at least three main contributions of our work beyond the proposal presented in Ref [5]:

- 1.) We studied a real and practical physical system where linear and nonlinear wave behavior is naturally present and the transition between them is straightforward. Hughes et al. did not present a practical physical system and thus their model can be considered as purely theoretical. We believe that our work consists in this sense much more than a ‘simple physical realization’, as the underlying nonlinear dynamics is fundamentally different from the model presented in Ref. 5, and thus it is not trivial that the same principles are applicable for both nonlinear systems. Our system has immediate practical implications and based on the rich literature of spin waves and high accuracy of the micromagnetic model it is possible to give preliminary metrics of a system based on the proposed framework. The model system we designed is practically realizable using widely available fabrication techniques and in principle enables on-chip integration and direct processing of microwave signals (e.g.: in telecommunication).

- 2.) We showed that computational performance of wave interference can be significantly increased by introducing nonlinear behavior. This was hypothesized by Hughes et al., but their simulations did not support this statement (linear operation achieved slightly better performance in their simulations, the opposite of the expectation). Our work showed improved performance using moderate levels on nonlinearity when the training was performed for the same task and using the same data set as in Ref 5, proving that we could successfully exploit nonlinearity in our simulations. In addition, we demonstrated a test problem which is not solvable using only linear wave interference. Our simulations showed that the training algorithm was unable to train the device to perform this function using only linear wave dynamics, but the training achieved very good results after a threshold of nonlinearity was surpassed (by increasing the excitation amplitude). We also showed that in the spin-wave-based system too strong nonlinearity leads to degraded performance due to chaotic behavior (the result of modulational instability in spin waves).
- 3.) Perhaps most importantly, our work paves the way for a new class of spin-wave-based (magnonic) devices, presenting also a design methodology that is likely useful for a large class of neuromorphic, microwave signal processing, possibly logic devices. We do not propose a simple physical illustration for wave-based computing concepts – we present a toolbox to design a new class of spin-wave devices. So far all spin-wave-based devices in the literature were designed manually, automatized inverse design of components for spin waves was not demonstrated, probably due to the high complexity of the underlying physics. As we also mention in the introduction, our paper is not viewed best as a physical illustration of wave-based computing concepts, but rather, as a design and a design toolbox for a large class of magnonic devices.

Here is how we incorporated this discussion in the revised version:

“More broadly, we introduce a new class of spin-wave-based computing devices (which are also referred to as magnonic devices) that can perform neuromorphic, signal-processing and logic operations using nonlinear wave interference. We demonstrate machine learning as a tool to design a variety of such devices. Magnonic devices have already been experimentally demonstrated for logic, signal processing and optically inspired computing^{2,3} and widely regarded as a promising beyond-Moore computing paradigm. The methodology we present here paves the way for straightforward (almost automatic) design for so far unexplored spin-wave circuits.”

“Our work also addresses the challenge of device integration, possibly the biggest obstacle in the way of practically useful spin-wave computing. There is a number of spin-wave-computing devices designed in the literature, but it is challenging to interconnect such stand-alone magnonic components to functional processor [7]. Our approach circumvents the difficulty of wiring individual nanoscale spin-wave devices: the device and the interconnect are indistinguishable in the wave-computing substrate and they are also designed as one.”

Some other comments:

1) The output is defined as time-integrated wave intensity over certain area. Is the intensity defined as the square of the spin wave amplitude? If so, how to pick up such time-integrated signal in reality? And how crucial is the size of the area?

Yes, the output is defined as the square of the dynamic magnetization components summed over the area of the detector. The detector area can be any size in the simulation, the training algorithm can successfully train with single-cell detectors or if the detector size covers multiple wavelengths. In reality, the detector size typically needs to be matched to the spin-wave wavelength if the detector is sensitive to the magnetization/field component directly. However, inverse spin hall effect is often used for detection of spin waves, which produces a DC signal proportional to the intensity of the spin wave, and the detector size can be much larger than the wavelength.

We would like to point out that the choice of time-integrated output is rather arbitrary, and the training algorithm could allow other methods of readout if that is more appropriate for the specific physical system.

Admittedly, energy efficient readout is the Achilles heel of magnonic devices – the small read-out areas generate relatively small signals that have to be picked up by amplifiers. Most low-power nanoelectronic devices face a similar challenge when transduced to electrical signals.

Here is how we incorporated this discussion in the revised version:

“It is worth noting that the computational learning engine can implement any kind of readout mechanism, thus any shape, size and type of detector may be used, depending on the application. For example, inverse spin-Hall effect [20] (iSHE) might be used to detect directly the amplitude of spin waves, while the detector size is not limited by the spin-wave wavelength.”

[18] Breikreutz-von Gamm, Stephan, Adam Papp, Eugen Egel, Christian Meier, Cenk Yilmaz, Leonhard Heiss, Wolfgang Porod, and Gyorgy Csaba. "Design of on-chip readout circuitry for spin-wave devices." IEEE Magnetics Letters 8 (2016):1-4.

[19] Chumak, A. V., et al. "Direct detection of magnon spin transport by the inverse spin Hall effect." Applied Physics Letters 100.8 (2012): 08240

2) The magnetic dots with perpendicular anisotropy placed on top of the magnetic YIG film served as trainable weights. During the simulation, I assume the authors only simulated the magnetic dynamics within the YIG film and the magnetic dots are assumed to be static. But in reality, the magnetization in the magnetic dots would also have dynamics. Will or will not the dynamic motion in the magnetic dots alter the results, and why?

The Reviewer correctly described the simulated scenario and the static assumption. We estimate the dynamic field component to be orders of magnitudes lower than the static stray field, placing this effect below the noise level. There are multiple reasons suggesting this: YIG has an order of magnitude lower saturation magnetization than metallic ferromagnets, thus its stray field is comparably smaller. Spin-wave amplitudes represent only a few percent change of the magnetization even at nonlinear signal levels. Thus, the dynamic field component acting on the nanomagnets is well below mT levels, which dwarfs in comparison with hundreds of mT internal PMA fields. Also, Co|Pt multilayered material systems tend to have very high damping, thus any dynamic excitation dies out quickly.

We now mention this argument in the paper and cite one of our paper where this is discussed in more detail. Please also refer to our answer to question 5 of Reviewer 1 above, where the updated article text is also shown.

3) Fig. 3 shows the intensity of the wave have significant influence on the results, i.e. the non-linear spin wave enhances the vowel-recognition rate. What's the principle behind such enhancement?

We believe Ref. 5 also addressed this question by drawing an analogy between RNNs and wave interference. In Section 5.4 we attempted to show the equivalence of a linear spin-wave scatterer and a perceptron layer (without the activation functions at the outputs). Linear wave interference is equivalent to linear transformations, thus training a linear scatterer can only learn linear transformations (i.e., multiplication by a matrix). This obviously has limited use for computation, and single perceptron layers are known to be limited to linearly separable classification problems. (We think the vowel recognition problem chosen by Hughes et al. is approximately a linearly separable problem, since the vowels can be separated based on their specific spectral components with good accuracy.) Staying with the neural network analogy, nonlinearity in the wave dynamics can provide a ‘distributed activation function’ within the scatterer, thus surpassing the single perceptron layer, and making it akin to deep neural networks. Besides this, waves have a finite propagation speed and can be scattered backward in the scatterer, making it possible for loops to arise, and also providing a type of fading memory in the system. These properties make a nonlinear scatterer similar to an RNN, and thus we expect that this physical system could be used to tackle computing problems where RNNs were successful.

We tried to make this point more explicit in the revised version:

“Compared to the linear operation, nonlinear wave propagation implements a ‘distributed activation function’, thus the spin-wave substrate is more akin to multilayer neural networks. Moreover, due to back-scattering of spin waves, loops can form in the scatterer that resemble the operation of RNNs. Due to the delay of wave propagation such loops could also implement a fading memory, which is not exploited in the current investigation, but provide additional potential for the spin-wave substrate.”

Reviewer 3

The authors numerically study the propagation of spin waves in a scattering medium. The authors train the scattering medium to perform some computations. Specifically, they focus on frequency-dependent scattering to analyze vowel data, including nonlinearity in the evolution.

The results are not particularly surprising, given the many recent papers on computing by waves (e.g., PRX 8, 041037 (2018)). For this reason, I am not supporting the acceptance by NCOMM.

The main drawback is the absence of experimental results, which may eventually boost the impact of the paper. It is well established that waves compute by proper training of the medium, irrespectively of the type of waves (as outlined in ref. 5). The fact the authors report some simulations with spin waves is not unexpected or exciting.

Thank you for bringing this paper to our attention (the paper is cited now) and please refer to our introductory comments where we argue for the significance of our paper in the context of magnonic devices. While the mentioned wave-computing paper is interesting and relevant, the WIFI signals by no means represent a practical on-chip realizable computing device. In the cited paper, the nanophotonic implementations are also studied in the linear regime. Our paper shows the utility of the nonlinear wave-computing paradigm in a practical, on-chip implementation, which is a kind of implementation that is researched by a large community of researchers. Our paper’s primary goal is to advance the field of magnonics and not only to serve as an illustration of wave-computing concepts. We agree with the reviewer that the results may not be very surprising from a fundamental point of view, but our goal was to show realistic device applications for nonlinear waves and in this respect, we believe our work is a first.

Here is how we incorporated this discussion in the revised version:

“We finally note that our paper is not the first to describe physically realizable wave-computing systems: there are other proposals (for example³²) that even experimentally demonstrate wave-based computing protocols. But, to our knowledge our paper is the first to show an on-chip, integration-friendly wave computing device where nonlinearities play a key role.”

“The techniques we demonstrate pave the way to magnonic (spin-wave) devices way beyond the complexity of stand-alone logic gates or elementary signal processors.”

See also our answers to Reviewer 2 and the introductory remarks. We put our work in context within magnonics research community.

Another major drawback is that the authors do not test their simulation with a standard dataset for benchmarking machine learning methods, as the MNIST.

Hence, I am against the acceptance of the manuscript.

Please see our arguments to Reviewer 2. Since we design a complex nonlinear scatterer without any approximations (beyond what is included in micromagnetics theory), the size of the designed system is

limited and thus the complexity of the devices designed is relatively low – for this reason, we used a relatively simple dataset for testing. We believe that our system is significantly more complex than most nanodevice proposals that are studied via physical simulations – but still too small to compare them to large-scale, software-implemented networks.

Moreover, we used the same dataset as Ref [5], as we think direct comparison to their results could be interesting, most importantly because our work clearly demonstrates the use and benefits of nonlinearity – an important result that Ref [5] was apparently lacking. We improved on the description of the dataset, as also requested by Reviewer 1. Please see our remarks for the Reviewer 1.

Minor comments follow

- I do not like the general style of the paper; many vague statements and colloquial and verbose style ("skyrocketed," "stellar numbers," "soft-nonlinear," "is less of a problem," among others)

We modified the listed expressions.

Here is how we incorporated this discussion in the revised version:

"The interest in neuromorphic computing hardware **increased significantly** ~~skyrocketed~~ in recent years..."

"These **estimations represent orders of magnitude improvement** ~~are stellar numbers~~ when compared to the above-mentioned energy of approximate convolution or floating point operation..."

"We used excitation amplitudes in the linear (1 mT) and **moderately soft** ~~soft~~-nonlinear (20 mT and 50 mT) regimes..."

"Transduction on the input side (creation of spin waves) is less **challenging** ~~of a problem~~ as can be done by acceptable efficiency using coplanar waveguides..."

- The authors vaguely remark the rule of nonlinearity. The may outcome is that the nonlinearity improves training. This conclusion is vague, not only because the training and testing datasets are absent but also because the authors do not discuss the model universality. Non-polynomial nonlinearity is a needed ingredient for universality, and no insight is given in this direction.

The basis of the computational model is described in Ref. 5, most of these arguments are valid for the spin-wave system. The data sets are publicly available, see Ref 5 (we used the same data set so the results would be comparable).

The normalized magnetization components are bounded in the [-1, 1] interval, thus, considering a multilayer neural network analogy, a bounded activation function is present in the system, implying universality. Showing that universality is practically achievable in the proposed system goes beyond the scope of the current investigation, but we believe we demonstrated that nonlinearity is practically exploitable in the proposed system.

"We briefly note that the activation function in the system is bounded in the [-1,1] interval, which is shown to be a satisfactory condition in multilayer feedforward networks for universality [34]. Although investigation of the limits of the computational capability of the spin-wave substrate is out of the scope of this paper, in principle, the presented substrate could approximate arbitrary functions."

Reviewers' Comments:

Reviewer #1:

Remarks to the Author:

This paper presents a numerical study of spin wave interference which is used to design a neural network.

There are many similar results in this field by using spin wave nonlinearity to process information for potential neuromorphic application. This work lacks novelty to meet the standard of Nature Communications. The main novelty of this work is not straightforward, and the contribution to the field of magnonics and neuromorphic computing is rather secondary to the previous research (such as Ref. 5, 8 etc). There is no experimental support of the main conclusions. The results do not have the novelty and major advances that would be expected from a Nature Communication paper. I recommend that the authors improve the paper by taking the three reviewers' suggestions into account and submit to a more specialized journal.

Reviewer #2:

Remarks to the Author:

With the revision and the reply letter, I believe that the authors have address all of my and the other two reviewer's questions in a satisfactory manner. Therefore, I would recommend the paper to be published on Nature Comm.

Reviewer #3:

Remarks to the Author:

I think that the authors provide vague and nonfactual answers to my criticisms. I confirm my opinion against the acceptance of the manuscript by nature communications.

General remarks to all the reviews:

We would like thank you for the opportunity for the present additional round of review (appeal) of our paper. Per the suggestion of the Editor, this answer letter contains our response to the first round of review from reviewer one, along with the response to the additional questions raised in the second round of review. We also comment on the third reviewer's original opinion, as his/her second review contains no additional information compared to the first one.

Please find below our responses to all the points raised by the reviewers. We copied their comments (shown in green), our replies are shown in black, and we show revisions in blue, as they appear in our revised manuscript. A marked-up copy of the manuscript is also attached, which shows the differences between the first revision and the present (second revision) of our paper.

As a general remark: in this response, we emphasize that our paper makes a contribution to two different research areas.

The first and main significant contribution of our paper is that it presents the design and analysis of a new, highly realizable magnonic device that is capable of neuromorphic functionality based on non-linear spin-wave interference. In other words, our work presents a realistic and realizable physical system as a wave-based computational platform, and it is not simply at the level of abstract non-linear wave equations as commonly found in the literature. Magnonics is at the forefront of 'beyond Moore' electronics devices, but so far, most devices have relatively low complexity. Our work addresses, and we believe solves, a much sought-after problem in magnonic device research. While stand-alone devices (logic gates, signal processors) have been intensely researched by the community, interconnecting these units into functional components remains a challenge. In our paper, we demonstrate the physics-based design of complex processing units that do not require additional interconnections or auxiliary electrical components (besides inputs and outputs) to function. In other words, our approach circumvents this problem by designing the device and the interconnections at once, realizing the desired input-output scenario. Our device is described by a highly predictive, full micromagnetic model that has proven highly successful and which is the standard in the micromagnetics community. We also cite recent experimental results to emphasize the realizability of our device.

The second, and perhaps less significant, contribution is that our work demonstrates the use of machine learning methods on a particular wave equation and shows how such a wave-propagation medium can be turned into a neuromorphic computing platform. We are well aware of the fact that the computing potential of non-linear waves has been and is being explored by a number of groups, and this aspect of our work, while still useful, does not break new ground.

Reviewer 1 - comments to the original (first) submission

In this work, the authors propose a spin wave-based neural network, which can perform various neuromorphic computing functions, such as spectrum analyzer and vowel recognition. Their results show that such-designed device can indeed exploit nonlinear waves to achieve better performance in problems beyond linear signal processing. The idea of using the propagation and interference of spin wave for neuromorphic computing is interesting and novel, and this topic is promising for a new class of analog machine learning platforms. In general, the manuscript can be accepted for publication after the following issues are clarified and corrected.

1) Fig. 3 demonstrates the cross-entropy loss of training sets, the confusion matrices for the testing set in linear and nonlinear cases, respectively. What about the cross-entropy loss of testing sets and the confusion matrices of the training sets? Most importantly, the prediction accuracy as a function of training epochs over the testing and training datasets should be given as shown in [Tyler W. Hughes et al., *Sci. Adv.* 5, 6946 (2019)]. The authors claim that the confusion matrices on the training data set are perfectly diagonal for all amplitudes due to the small training sets and the comparably large internal complexity of the scatterer. It might be true to some extent. In my opinion, it is necessary to show above mentioned elements to draw more solid conclusions and prove that there is no overfitting problem. In addition, please give more details of the training dataset and testing dataset in one paragraph. Ref. [5] (*Science Advances* 5, no. 12 (2019): eaay6946.) in their references clearly explained the training details as follows. Please follow their example to give a more clear description.

We updated Fig. 3. according to the suggestions of the Reviewer, adding new data from new simulations. Figure 3c now includes the loss during the training epochs for both the training and testing data sets in case of both the linear and non-linear operation. Similarly, we calculated prediction accuracy for every training epoch and plotted the test/train accuracy for both the linear and non-linear cases (Fig. 3d). The confusion matrices for the training data set are perfectly diagonal (as we indicated in the text before), and this fact is clearly visible now in the accuracy of the curves (in Fig. 3d, the accuracy reaches 100 % very quickly on the training dataset for both the linear and non-linear cases). In light of this, we did not display the perfectly diagonal confusion matrices for the training sets, since they do not contain additional information.

We now included more information about the data set. We also emphasize that we used the same data set as Ref [5] for better comparability, and thus we also refer to their description of the data set.

Here is how we incorporated this discussion in the first revised version:

Updated Fig 3. and figure captions (see in the manuscript).

“For comparability, we used the same samples of the dataset as [5] but using only the male samples (due to computational resource limitations). In the dataset vowels ‘ae’, ‘ei’, and ‘iy’ were recorded inside the words ‘had’, ‘hayed’, and ‘heed’, we cropped these samples and only used the middle part of the waveform where the vowel was audible.

(...)

We used two or four samples of each vowel as a training set. The rest of the 45 samples for each vowel was used as test set.”

2) In section 2.3, they think the operation in the nonlinear regime achieved good results, with the highest amplitude excitation giving the best outcome. Is it possible to plot the relationship between the outcome (such as the final loss and the prediction accuracy) and the excitation amplitude? Is there a saturation point? Or is there a maximum value limit for the excitation amplitude?

We performed additional simulations on the vowel recognition device with multiple excitation amplitudes. The accuracy of the device on the testing set as a function of excitation amplitude is now depicted in Fig. 3g. We point out three separate amplitude regions: in the linear region, the performance of the device is independent of the amplitude. As the amplitude increases into the nonlinear region (after a sudden transition) the performance of the device improves significantly. Further increase of the amplitude gradually hinders the performance of the device, which we attribute to the onset of chaotic behavior of spin waves (modulational instability). We believe these three regions are fundamental to spin waves and similar behavior can be expected for other training tasks as well (as in case of the one described in Section 2.3).

We now added a paragraph about these results at the end of Section 2.2.

Here is how we incorporated this discussion in the first revised version:

“We also performed simulations using multiple excitation amplitudes and plotted the corresponding accuracy achieved on the testing data set (Fig. 3g). We observe three distinct regions depending on the amplitude: linear, nonlinear, and chaotic (strongly nonlinear) region. In the linear regime the accuracy of the system is modest, and independent of the amplitude. When the amplitude reaches a certain threshold level, the recognition accuracy suddenly improves significantly, and reaches maximum performance around 50~mT excitation field. We attribute this improvement to the emergence of nonlinear effects. However, further increase of the amplitude does not yield to better accuracy due to increasingly chaotic spin-wave dynamics, and the system accuracy eventually degrades even below the accuracy of the linear operation.”

3) The authors mention that the linear and nonlinear wave equations several times. Is it possible to provide the corresponding theoretical formulas or representative references, which can give readers a clearer physical scenario.

We now included references in Section 5.2 that give a better description of the physical scenario. We would like to point out that the nonlinear behavior of spin waves is more complicated to handle analytically than the nonlinear wave equation considered in Ref [5], thus we have to completely rely on numerical methods.

We also added a brief explanation about the physical background of the nonlinearity: magnetization dynamics depends on the local magnetic field, but the local magnetic fields depend also on the amplitude of the magnetization oscillations. We do not state the full system of equations, but mention this point in the Methods section.

Here is how we incorporated this discussion in the first revised version:

“A detailed description of the dispersion relation and nonlinearity of spin waves can be found in [30]. The nonlinear wave equation for spin waves cannot be stated in a simple form as the source of nonlinearity is the amplitude dependent magnetic field, so the nonlinearity enters to the dynamic equations via Maxwell's equations. Experimental demonstration of chaotic behavior of spin waves due to modulational instability at very high amplitude levels can be found in [31]. In this work we used moderate amplitudes to avoid chaotic behavior, which degrades the ability of the device to perform the desired functionality.”

[30] Wu, Mingzhong. "Nonlinear spin waves in magnetic film feedback rings." *Solid State Physics* 62 (2010): 163-224.

[31] Wu, Mingzhong, et al. "Excitation of chaotic spin waves through modulational instability." *Physical review letters* 102.23(2009): 237203

4) To improve the significance of this work, I would like the authors to make a comparison between the proposed device with the conventional RNN in terms of the prediction accuracy.

The focus of our study is the investigation of a physical substrate that could perform computing tasks, and the design (training) of such a substrate. Our primary goal was to show that a spin-wave scatterer can perform computing tasks and nonlinear scattering increases the complexity of the computing problems. We are simulating a small system with all the micromagnetic interactions accurately modeled, which is a computationally intensive task. Benchmarking such a simulated physical system against an artificial neural network that is directly realized in software would, in our opinion, not be a meaningful and fair comparison. Our model system is very small compared to most systems discussed in the literature of Artificial Neural Networks. The reason for this is that is the amount of computational resources it requires, since we need to simulate a complex physical system. This places a limitation on the complexity of the problems we can currently tackle, and also the size of the training and testing data sets. Of course, benchmarking a physically realized system would be a meaningful comparison, as the physical substrate would operate real-time on the data, without the simulation overhead.

We used the vowel recognition problem to be comparable to the results of Ref [5], and we believe that we succeeded in demonstrating that nonlinear spin-wave behavior has greater computational complexity than what is possible using linear wave interference (in this respect the simulation results of Ref [5] were inconclusive). Before attempting a meaningful benchmark against well-established methods, we need to scale up the system to be able to solve problems with practical significance. We don't see any fundamental limitation in this sense, but the efficiency of our training algorithm needs to be improved so that we could apply it to larger-scale problems. We also plan to work on experimental verification of the presented principles, which would allow us to design a more realistic system, and better estimation on the limitations of the device.

We can more confidently make a comparison between electrical and spin-wave hardware. The spin-wave interference formation process takes a couple of nanoseconds in time and the size of our entire computing substrate is 10 micrometers – a digital or analog electronic circuitry performing the same RNN task would likely consist of hundreds (possibly thousands) of transistors, with significantly higher power consumption, significant routing complexity and higher footprint. These points are mentioned in the benchmarks section.

Here is how we incorporated this discussion in the first revised version (just before concluding the paper):

“The computational learning method we use is computationally intensive (see Section 5.1) and we could not design larger systems than the ones presented here and compare those to state of art neural networks.”

5) In the main text, the authors claim that the programming nanomagnets assumed to exhibit strong perpendicular magnetic anisotropy (PMA). However, in section 5.2 (e.g. methods), the effective magnetic field is just described by the sum of the external field, the dipole field and exchange interaction. I wonder why the PMA field is missing here.

The magnetic system consists of two parts: a YIG thin film and Co|Pt nanomagnets on top. PMA is present in Co|Pt (and similar) multilayer systems, but not in YIG. In the simulations we assume that PMA keeps the nanomagnets magnetized completely in out-of-plane direction, and the static stray field of such magnets is possible to calculate analytically. Thus, in the dynamic micromagnetic simulations the PMA magnets themselves are not present, only their stray fields are included.

To make things clearer, we added a sentence stating that physically the ‘programming’ field is included in an external field component:

“The external field contains a bias field, the time-dependent excitation field, and also the field coming from the PMA magnets (which is optimized by machine learning). We neglect

any dynamics of the PMA magnets (e.g. due to coupling of spin waves from YIG) as these fields are orders of magnitude lower [9].”

Reviewer 1 - second review.

REASON FOR APPEAL: This review completely ignores the revisions we had made in response to this reviewer’s original comments. Instead, this reviewer now raises additional points (unrelated to the first review), which we address and refute below.

This paper presents a numerical study of spin wave interference which is used to design a neural network.

There are many similar results in this field by using spin wave nonlinearity to process information for potential neuromorphic application. This work lacks novelty to meet the standard of Nature Communications. The main novelty of this work is not straightforward, and the contribution to the field of magnonics and neuromorphic computing is rather secondary to the previous research (such as Ref. 5, 8 etc). There is no experimental support of the main conclusions. The results do not have the novelty and major advances that would be expected from a Nature Communication paper. I recommend that the authors improve the paper by taking the three reviewers' suggestions into account and submit to a more specialized journal.

As already stated above, the above comments of the reviewer do not address our revision and the changes we had made to the paper in response to his/her comments.

In addition, most of the above statements are not correct, which we address below:

1: The statement “... many similar results in this field by using spin wave nonlinearity to process information for potential neuromorphic application.” is factually wrong. Indeed, here are papers using linear spin wave effects¹, there are works using caustics (anisotropic propagation)², and there are RF devices and directional couplers that use a non-linearity (but these are Boolean devices)³. We are not aware of any papers that use

¹ Papp, Ádám, Wolfgang Porod, Árpád I. Csurgay, and György Csaba. "Nanoscale spectrum analyzer based on spin-wave interference." *Scientific reports* 7, no. 1 (2017): 1-9.

Gertz, Frederick, Alexander Kozhevnikov, Yuri Filimonov, and Alexander Khitun. "Magnonic holographic memory." *IEEE Transactions on Magnetics* 51, no. 4 (2014): 1-5.

Chumak, Andrii V., Vitaliy I. Vasyuchka, Alexander A. Serga, and Burkard Hillebrands. "Magnon spintronics." *Nature Physics* 11, no. 6 (2015): 453-461.

Barman, A., Gianluca Gubbiotti, Sam Ladak, Adekunle Olusola Adeyeye, Maciej Krawczyk, Joachim Gräfe, Christoph Adelman et al. "The 2021 magnonics roadmap." *Journal of Physics: Condensed Matter* (2021).

² Heussner, Frank, Giacomo Talmelli, Moritz Geilen, Björn Heinz, Thomas Brächer, Thomas Meyer, Florin Ciubotaru et al. "Experimental realization of a passive gigahertz frequency-division demultiplexer for magnonic logic networks." *physica status solidi (RRL)–Rapid Research Letters* 14, no. 4 (2020): 1900695.

³ Wang, Qi, Philipp Pirro, Roman Verba, Andrei Slavin, Burkard Hillebrands, and Andrii V. Chumak. "Reconfigurable nanoscale spin-wave directional coupler." *Science advances* 4, no. 1 (2018): e1701517.

non-linear interference to perform neuromorphic computation. Also, we are not aware of works that connect multiple neuron-like devices for enhanced complexity.

2. The statement "... the contribution to the field of magnonics and neuromorphic computing is rather secondary to the previous research (such as Ref. 5, 8 etc)." is not correct. We clearly describe that [5] serves as a starting point of research, but [5] uses a wave equation that does not describe a specific physical system and does not prove the critical role of non-linearity. In other words, [5] is not describing any realizable electronic device, making it significantly different from our paper. Regarding [8], it is clearly stated in our paper that [8] and our paper were submitted simultaneously. In fact [8] refers to the preprint (arxiv) version of our paper, and we had coordinated our original submission with the authors of [8]. We believe that [5] and [8] and our present paper together break new ground to the design of computing devices and by no means lessen each other's significance. As such, the existence of [5] and [8] should not be a reason to reject our paper, but rather a reason for its acceptance.

3. The statement "There is no experimental support of the main conclusions." is not correct. While [5] and [8] are pure theory papers without any experimental support, there is recent experimental evidence of our micromagnetic computing platform, and we now include this reference in the newly revised version.

We added a few remarks / references to avoid potential misinterpretation of our results. We emphasize that the complexity of the device function is equivalent to that of interconnected layers of neurons and therefore solves the challenging device integration problem in magnonics. We also refer to a recently published experimental magnonics paper (where we are co-authors) in order to emphasize the predictive power of micromagnetic simulations.

The new references are related to spin-wave devices:

Mahmoud, Abdulqader, Florin Ciubotaru, Frederic Vanderveken, Andrii V. Chumak, Said Hamdioui, Christoph Adelman, and Sorin Cotofana. "Introduction to spin wave computing." *Journal of Applied Physics* 128, no. 16 (2020): 161101.

Grollier, Julie, Damien Querlioz, K. Y. Camsari, Karin Everschor-Sitte, Shunsuke Fukami, and Mark D. Stiles. "Neuromorphic spintronics." *Nature electronics* 3, no. 7 (2020): 360-370.

New references related to experiments and the underscoring the predictive power of micromagnetics:

Papp, Ádám, Martina Kiechle, Simon Mendisch, Valentin Ahrens, Levent Sahin, Lukas Seitner, Wolfgang Porod, Gyorgy Csaba, and Markus Becherer. "Experimental demonstration of a concave grating for spin waves in the Rowland arrangement." *Scientific Reports* 11, no. 1 (2021): 1-8.

The text added to the paper:

Specifically, we design a nanoscale device that performs the functions of a multilayer neural network and which device is physically realizable using microelectronics-

compatible technologies. Our work builds on prior art in wave-based computing methods, but rather than choosing an abstract wave equation as its starting point, we start from the physics of magnetic materials. We show that the rich and complex physics of spin waves in a ferrimagnetic thin film can be engineered to perform neuromorphic computation.

Micromagnetic simulations in general, give a highly accurate and predictive description of magnetic behavior, without requiring fitting parameters. To give a recent example [Papp et. al. Scir Rep 11, 2021] shows how complex interference patterns and nonlinear excitations in a magnetic thin film can be engineered and subsequently measured experimentally.

Reviewer #2 (Remarks to the Author):

With the revision and the reply letter, I believe that the authors have address all of my and the other two reviewer's questions in a satisfactory manner. Therefore, I would recommend the paper to be published on Nature Comm.

We would like to thank this reviewer for the positive evaluation of our work! We are also grateful for the reviewer for his comments in the first round of revisions, which helped to greatly improve our paper.

Reviewer #3

I think that the authors provide vague and nonfactual answers to my criticisms. I confirm my opinion against the acceptance of the manuscript by nature communications.

Since this assessment lacks detail, we would like to use this opportunity to elaborate on our answers to some points for the first round of review.

Reviewer 3 - original opinion (excerpt)

The authors numerically study the propagation of spin waves in a scattering medium. The authors train the scattering medium to perform some computations. Specifically, they focus on frequency-dependent scattering to analyze vowel data, including nonlinearity in the evolution.

The results are not particularly surprising, given the many recent papers on computing by waves (e.g., PRX 8, 041037 (2018)). For this reason, I am not supporting the acceptance by NCOMM.

The main drawback is the absence of experimental results, which may eventually boost the impact of the paper. It is well established that waves compute by proper training of the medium, irrespectively of the type of waves (as outlined in ref. 5). The fact the authors report some simulations with spin waves is not unexpected or exciting.

We believe that the above opinion stems from misunderstanding and factual errors. Our paper describes a device and physical (device) simulations. It was not the intent of our paper to add to ' many recent papers on computing by waves ', which are overwhelmingly mathematical works. This includes ' PRX 8, 041037 ', which is an interesting work, but has little to no relevance to electronic device design. It is a misleading oversimplification that ref [5] states that ' waves compute by proper training of the medium, irrespectively of the type of waves ', because ref [5] assumes a specific type of non-linear wave equation and draws an analogy between that wave equation and a recurrent neural network.

Let us also point out that ' some simulations ' does not do justice to our complete micromagnetic simulator with an FFT/convolution based magnetostatic field solver, which includes exchange and anisotropy magnetic fields, magnetic field calculation of a waveguide, time-dependent solution of precession equations (the LLG equations) on a large grid, and all this solution implemented in a such a way that auto gradients (backpropagation-based machine learning) can be implemented in designing a scattering field. We are not aware of any micromagnetic simulation tool with such computational capability and complexity, and we implemented such a complex tool for exactly the reason that we design a realizable electronic device.

The reviewer's statement 'The fact the authors report some simulations with spin waves is not unexpected or exciting.' seems to indicate that the reviewer sees our work as an illustration, or numerical example, for training a wave equation, which is a misunderstanding. Let us make the point that just as photonic device design is not an unexciting exercise of solving Maxwell's equations, our device and device design method is not an exercise on demonstrating computational power of idealized wave equations.

In order to avoid potential misunderstanding we emphasize the device aspects of our work and we make a connection to experimental work by our group. We hope that this emphasizes the feasibility, experimental realizability, and the practical nature of our device.

The changes in the paper are the same as given in the response for reviewer 1:

The new references are related to spin-wave devices:

Mahmoud, Abdulqader, Florin Ciubotaru, Frederic Vanderveken, Andrii V. Chumak, Said Hamdioui, Christoph Adelman, and Sorin Cotofana. "Introduction to spin wave computing." *Journal of Applied Physics* 128, no. 16 (2020): 161101.

Grollier, Julie, Damien Querlioz, K. Y. Camsari, Karin Everschor-Sitte, Shunsuke Fukami, and Mark D. Stiles. "Neuromorphic spintronics." Nature electronics 3, no. 7 (2020): 360-370.

New references related to experiments and the underscoring the predictive power of micromagnetics:

Papp, Ádám, Martina Kiechle, Simon Mendisch, Valentin Ahrens, Levent Sahin, Lukas Seitner, Wolfgang Porod, Gyorgy Csaba, and Markus Becherer. "Experimental demonstration of a concave grating for spin waves in the Rowland arrangement." Scientific Reports 11, no. 1 (2021): 1-8.

The text added to the paper:

Specifically, we design a nanoscale device that performs the functions of a multilayer neural network and which device is physically realizable using microelectronics-compatible technologies. Our work builds on prior art in wave-based computing methods, but rather than choosing an abstract wave equation as its starting point, we start from the physics of magnetic materials. We show that the rich and complex physics of spin waves in a ferrimagnetic thin film can be engineered to perform neuromorphic computation.

Micromagnetic simulations in general, give a highly accurate and predictive description of magnetic behavior, without requiring fitting parameters. To give a recent example [Papp et. al. Scir Rep 11, 2021] shows how complex interference patterns and nonlinear excitations in a magnetic thin film can be engineered and subsequently measured experimentally.

Reviewers' Comments:

Reviewer #1:

None

Reviewer #2:

Remarks to the Author:

For author's reply to the second review of Reviewer 1, I think the authors have made valid rebuttal that the three statements of the reviewer 1 are not based on solid facts. Therefore, I regard the second review of Reviewer 1 does not provide fair assessment.

The main objection from Reviewer 3 is the similarity of this manuscript and previous works on computing by waves. This criticism seems to be too vague. Computing by waves is a general concept, and it can have very different scenarios depending on the types of waves and medium as the authors pointed out.

In conclusion, I believe the authors have addressed all all technical concerns in a satisfactory manner and I support the paper for publication on NCOMM.